# Non-Collagenous Dentin Protein Binding Sites Control Mineral Formation during the Biomineralisation Process in Radicular Dentin

**DOI:** 10.3390/ma13051053

**Published:** 2020-02-27

**Authors:** Cristina Retana-Lobo, Juliane Maria Guerreiro-Tanomaru, Mario Tanomaru-Filho, Beatriz Dulcineia Mendes de Souza, Jessie Reyes-Carmona

**Affiliations:** 1LICIFO—Laboratory of Research in Dental Sciences, Department of Endodontics, Faculty of Dentistry, University of Costa Rica, 11502 SJO, Costa Rica; cristina.retana@ucr.ac.cr; 2Department of Restorative Dentistry, São Paulo State University (UNESP), School of Dentistry, 14801385, Araraquara, SP, Brazil; jutanomaru@uol.com.br (J.M.G.-T.); tanomaru@uol.com.br (M.T.-F.); 3Department of Dentistry, Federal University of Santa Catarina, Florianopolis, 88040-900, Florianopolis, Santa Catarina, Brazil; dentbiadms@gmail.com

**Keywords:** biomineralisation, carbonate apatite, bioactive materials, dentin remineralization, matrix metalloproteinases, non-collagenous proteins, radicular dentin

## Abstract

The biomineralisation of radicular dentin involves complex molecular signalling. Providing evidence of protein binding sites for calcium ions and mineral precipitation is essential for a better understanding of the remineralisation process. This study aimed to evaluate the functional relationship of metalloproteinases (MMPs) and non-collagenous proteins (NCPs) with mineral initiation and maturation during the biomineralisation of radicular dentin. A standardized demineralisation procedure was performed to radicular dentin slices. Samples were remineralised in a PBS-bioactive material system for different periods of time. Assessments of ion exchange, Raman analysis, and energy dispersive X-ray analysis (EDAX) with a scanning electron microscope (SEM) were used to evaluate the remineralisation process. Immunohistochemistry and zymography were performed to analyse NCPs and MMPs expression. SEM evaluation showed that the mineral nucleation and growth occurs, exclusively, on the demineralised radicular dentin surface. Raman analysis of remineralised dentin showed intense peaks at 955 and 1063 cm^−1^, which can be attributed to carbonate apatite formation. Immunohistochemistry of demineralised samples revealed the presence of DMP1-CT, mainly in intratubular dentin, whereas DSPP in intratubular and intertubular dentin. DMP1-CT and DSPP binding sites control carbonate apatite nucleation and maturation guiding the remineralisation of radicular dentin.

## 1. Introduction

Dentin is the mineralised tissue underlying the enamel and cementum constituting the main component of the tooth. During dentinogenesis, odontoblasts synthesise collagenous and non-collagenous proteins (NCPs) and form an unmineralised extracellular matrix (ECM), in which type I collagen is the major constituent [1,2,3,4,5]. Collagen self-assembles into fibrils to form a passive scaffold that acts as a template for mineral deposition at specific sites in the biomineralisation process [4,6,7]. However, the role of the collagen matrix during nucleation of amorphous calcium phosphate and its transformation into oriented crystals of apatite remain controversial.

The controlled growth and formation of the mineral phase involve a sequential series of induction events [2,3,8,9]. Non-collagenous matrix acidic macromolecules could bind calcium ions, acting as nucleators or inhibitors of mineral formation from spontaneously precipitating solutions [10,11]. NCPs are believed to initiate and modulate the mineralisation of collagen fibres [5,12]. One particularly important category of NCPs is the SIBLING (small integrin-binding ligand, N-linked glycoprotein) family, which includes dentin sialophosphoprotein (DSPP), dentin matrix protein 1 (DMP1), bone sialoprotein, osteopontin, and matrix extracellular phosphoglycoprotein [5,12,13]. SIBLING family members have some common features, such as the presence of phosphorylation, glycosylation, and the Arg-Gly-Asp (RGD) cell-binding sequence as well as similarities in genomic organisation and localisation that are involved as key factors in the mineralisation of dentin and bone [12].

Further, the biomineralisation process requires active extracellular enzymatic control performed by different proteinases, predominantly members of the matrix metalloproteinase (MMP) family. MMPs are involved in physiological tissue development, tissue remodelling, and pathological processes [14]. Among endogenous MMPs, MMP-2, MMP-9, and MMP-20 have been isolated from mature human mineralised and demineralised dentin [15,16,17,18]. Cleavage by MMPs converts structural matrix proteins such as NCPs into active signalling molecules, for instance MMP-2 cleaves DMP1 and DSPP to release biologically active peptides [15].

DMP1 contains a large number of acidic domains, which may provide the molecular properties necessary to control the formation of oriented calcium phosphate crystals [5,19]. DMP1 is processed into three major components: an N-terminal (37 kDa) fragment, a highly phosphorylated C-terminal (57 kDa; DMP1-CT) fragment (both of which act as hydroxyapatite nucleators), and a proteoglycan form of the N-terminal fragment (known as DMP1-PG) [5,20,21]. In particular, DMP1-CT self-assembles to form a fibrillary template for the precipitation of hydroxyapatite crystals [2]. Additionally, DSPP is proteolytically processed into dentin sialoprotein (DSP), dentin phosphoprotein (DPP), and a proteoglycan form of DSP (DSP-PG), which originate from the N-terminal and C-terminal regions [22]. These variants are distributed differently among individual compartments of the tooth and play different roles during dentinogenesis. DMP1 and DSPP share similar tissue localisation and proteolytic processing [12,13,20,23,24]. Additionally, both have effective calcium-binding capacities and a high affinity for fibrillar collagen, which induces the heterogeneous nucleation of calcium phosphate crystals and regulates crystal growth, resulting in unique morphologies [3]. Certainly, cleaved DPP seems to play an essential role in the nucleation of calcium phosphate mineral [11], although its exact mechanisms of action are unknown.

Indeed, there is little consensus regarding the signalling molecules involved during biomineralisation. Apatite deposition among collagen fibrils is a key factor in the regeneration of mineralised tissues [25,26]. Controlled mineral nucleation on demineralised dentin involves complex molecular signalling that guides the site and rate of apatite formation [5]. In a previous study, we reported the ability of a bioactive material to biomineralise dentin in a phosphate-containing fluid [27]. We hypothesized that the recombinant DMP1 molecules were able to recognize a specific molecular signature on the apatite surface, thereby guiding calcium phosphate clusters during recruitment through the collagen matrix [27], and this special affinity could trigger an ion migration flux and attract crystal precipitation [2,5,8,10] in a process described as controlled biomineralisation [28]. However, providing experimental evidence for the role of NCPs and collagen in guiding mineral formation would require monitoring protein levels during the biomineralisation process.

From a clinical perspective, caries and acid erosion can cause dentin demineralisation. In view of this fact, coronal dentin biomimetic remineralisation strategies have been reported, which attempted tissue regeneration [8,9,29,30]. To our knowledge, those approaches succeed partially because most of these strategies achieved remineralisation using in vitro models and complex methodologies [4,9,29] that poorly reproduce the clinical environment; hence, the possibility of extrapolating the protocols is limited.

However, coronal and radicular dentin differs in their environment, structure, and in their organic and inorganic components. Coronal dentin exhibits significantly higher tubule density, whereas radicular dentin has larger diameter tubules and smaller peritubular cuff thickness, resulting in more peritubular dentin and a larger volume fraction of the matrix [31]. Differences in the presence of highly phosphorylated proteins (SIBLING family) between coronal and radicular dentin have also been described [32,33]. For example, DPP (DSPP fragment) levels in coronal dentin are approximately four times higher than those in radicular dentin in rodent incisors [33]. Differences in the structure and protein distribution between coronal and radicular dentin may indicate different substrates. Moreover, coronal dentin, when exposed, is surrounded by an environment with saliva and oral microbiota, whereas exposed radicular dentin is surrounded by cells in the periodontal ligament, bone, and tissular fluid with ions that provide an ideal environment to promote the biomineralisation process.

Due to its localisation in the tooth, radicular dentin can be exposed to multiple factors that promote dentin demineralisation, such as caries, erosion, pulp necrosis, infection of the root canal system, and diverse periodontal pathological conditions. Moreover, common chemicals used during endodontic procedures may cause collagen matrix degradation. To date, no reports have examined the remineralisation process of a radicular dentin matrix.

The aim of this study was to evaluate the functional relationship of MMPs (MMP-2 and MMP-9) and NCPs (DSPP and DMP1-CT) with the initiation and maturation of apatite, through a biomimetic remineralisation system of radicular dentin. The null hypothesis was that NCPs binding sites act as mineral nucleators guiding the remineralisation of radicular dentin collagen matrix with the aid of bioactive materials.

## 2. Materials and Methods


*Ethical Statement*


The research protocol was approved by the Ethics Committee of Universidad de Costa Rica (VR-467-2018).

### 2.1. Preparation of Human Radicular Dentin Specimens

Radicular dentin samples were prepared from 178 single-rooted extracted human teeth (Appendix A). The crowns and apical thirds of the roots were removed, and the middle root third was sectioned using a water-cooled low-speed ISOMET diamond saw (Buehler, Lake Bluff, NY, USA) to obtain 1065 dentin slices (2 mm H × 2 mm W × 2 mm L).

The dentin slices were treated according to Reyes-Carmona et al. [27] protocol. For SEM analyses, 190 samples remained untreated as a control group. Half of the remaining samples were partially (*n* = 380) or completely (*n* = 380) demineralised with 37% H_3_PO_4_ solution for 30 s and washed with distilled water. The samples were divided into five groups according their experimental treatment (*n* = 95), as described in Table 1, and in 19 subgroups (*n* = 5) according their experimental timepoint (1–15, 30, 60, 90, or 120 days).

For immunohistochemical analyses, 100 dentin slices were used and divided into the five treatment groups as shown in Table 1 (*n* = 25).

### 2.2. Biomimetic Remineralisation Model

Standardised Biodentine^®^ (Septodont, Saint-Maur-des-Fossés, Paris, France) discs (10-mm diameter) were prepared following the manufacturer´s recommendations. A calcium-free and magnesium-free Phosphate Buffered Saline (PBS) solution containing 136.4 mM NaCl, 2.7 mM KCl, 8.2 mM NaH_2_ PO_4_, and 1.25 mM KH_2_ PO_4_ in deionized water (pH 7.2) was prepared and filtered. Each sample and the bioactive disc were placed on opposite sides, distanced 15 mm apart, of a glass vial containing 20 mL of PBS for 4 months at 37 °C. Samples were collected according to the experimental times (1–15, 30, 60, 90, and 120 days).

### 2.3. Determination of pH and Calcium Ion Release

The PBS solution was collected in sterile specimen vials and replaced at 12 h, 1–15, 30, 60, 90, and 120 days to measure the pH and calcium ion release. After collection of the solution, the pH was determined with a pH metre (Orion Star A 221, Waltham, MA, USA), which was previously calibrated. Calcium ion release was measured using a Varian atomic absorption spectrophotometer (Spectra A220 Fast Sequential, Palo Alto, CA, USA). Data obtained were recorded and submitted to descriptive analysis.

### 2.4. Raman Analysis

Demineralised (*n* = 5) and remineralised (*n* = 5) dentin slices were subjected to Raman analysis to compare their chemical composition with control dentin (*n* = 5). Raman spectra were recorded using a Raman microspectrometer (ProRaman-L, Enwave Optronics Inc., Irvine, CA, USA). A 50× microscope objective (Leica Microsystems Inc., Buffalo Grove, IL, USA) was used, and the samples were excited using 45–50 mW of a 785 nm diode laser. Raman signal was collected in the spectral interval 800–1800 cm^−1^ and 1100–1800 cm^−1^. The integration time was 40 s, and spectral resolution was approximately 2 cm^−1^.

### 2.5. Composition and Ultrastructural Examination of Precipitates and Remineralisation Model

Samples prepared for SEM observation were mounted on an aluminium stub and sputter-coated with a 300-A° gold layer. The elemental composition of the precipitate phases in the sample surface was analysed by energy dispersive X-ray analysis (EDAX) with a scanning electron microscope (SEM) (S-570, HITACHI, Tokyo, Japan) at 15 kV. Three evaluations were performed for each sample in different areas. Serial SEM photomicrographs at different magnifications were taken to analyse the ultrastructure of the precipitate and the remineralisation process.

### 2.6. Immunohistochemical Analyses

All samples were processed using conventional histochemical techniques, embedded in paraffin, sectioned at 2-µm thickness, mounted on glass slides, and deparaffinised. The samples in each group were subdivided according the following primary antibodies and respective dilution ratios (*n* = 5): mouse monoclonal anti-DMP-1 CT (1:125 EMD Millipore, Burlington, MA, USA), rabbit polyclonal anti-DSPP (1:250 ABCAM, Cambridge, UK), rabbit polyclonal anti-MMP2 (1:600, Novus Biologicals, Centennial, CO, USA), and mouse polyclonal anti-MMP9 (1:200, Novus Biologicals, Centennial, CO, USA). Proteinase K (Dako Cytomation, Carpinteria, CA, USA) incubation was performed for antigen retrieval, following the manufacturer’s instructions. Nonspecific binding was blocked by incubating sections for 1 h with goat normal serum diluted in PBS. After overnight incubation at 4 °C with primary antibodies, the slides were washed with PBS and incubated with the conjugated secondary antibody SignalStain Boost detection reagent (Cell Signaling, Danvers, MA, USA) in a humidified chamber for 30 min at room temperature. Samples were washed, and visualisation was completed using the SignalStain DAB Substrate kit (Cell Signaling, Danvers, MA, USA) and counterstaining lightly with Mayer haematoxylin solution.

Images of the immunohistochemical stained tissue sections were acquired using a light microscope (Nikon, Eclipse Ti-5; Minato, Tokyo, Japan). Five images per sample were captured at 40× magnification. The threshold optical density was obtained using the NIH Image J 1.36b imaging software (National Institutes of Health, Bethesda, MD, USA). The total pixel intensity was determined, and data were expressed as optical density.

Data obtained were statistically analysed using GraphPad Prism 4 software (GraphPad Software Inc, San Diego, CA, USA). Analysis of variance (ANOVA) and the Tukey test were performed. A *p* ˂ 0.05 was considered to be statistically significant.

### 2.7. Zymography

Mineralised dentin powder was prepared from 25 single-rooted extracted human teeth. Roots and crowns were separated, and pulpal soft tissue and cementum were removed. Root dentin samples were reduced to fine powder by cryogenic grinding (MM 400, Retsch, Haan, Germany). Samples were stored at −80 °C until use.

Root dentin aliquots were divided into the following five groups—R1-Control: one lot of untreated dentin powder, R2: the dentin powder was demineralised with 37% H_3_PO_4_ for 30 s + distilled water, R3: as same as R2 + 5 min in 2% chlorhexidine + distilled water, R4: same as R2 + Biodentine, and R5: same as R3 + Biodentine.

The extraction of dentin proteins was performed via the Mazonni et al. [34] protocol, with minor modifications. Aliquots were subjected to electrophoresis under non-reducing conditions in a 5–20% gradient sodium dodecyl sulphate-polyacrylamide gel (SDS-PAGE) containing 1 mg/mL gelatine. Gels were washed and incubated overnight in activation solution (50 mmol/L Tris-HCl, 5 mmol/L CaCl_2_, pH 7.4). The gel was stained in 0.2% Coomassie Brilliant Blue R-250, destained in 10% acetic acid-10% methanol in H_2_O, and photographed using the ChemiDoc™ imaging System (Bio-Rad, Hercules, CA, USA) for further analysis.

## 3. Results

### 3.1. Assessment of Ion Exchange in Experimental Remineralisation Model

To assess ion release from the bioactive material and their subsequently incorporation into the precipitate, pH analyses (OH^−^ ion) and calcium ion release were measured in the PBS-solution collected from the remineralisation model. It was possible to observe a similar profile for all experimental groups (G2–G5); therefore, the mean value was used to describe the profiles through the experimental times.

The highest pH value was recorded during days 3 and 4 (pH 10.0–10.4) after placing the biomimetic remineralisation model. On days 5–9, the pH values remained at around 10 and subsequently declined, reaching pH 8.5 on day 30. The pH then remained stable (pH 8.4) until the end of the experiment (60–120 days). The pH values recorded for the PBS-solution associated with the control dentin slices remained at 7.2 throughout the experiment (Figure 1A).

Calcium ion release initially increased rapidly, and the highest amount of calcium release into the solution was recorded on day 3 (32.42 mg/L). Subsequently, calcium release decreased until day 15 and then remained stable (9.7–9.4 mg/L) until the end of the experiment (day 120). Trace amounts of calcium were detected in the solution associated with the control group, but these levels remained stable throughout the experiment (1.5–1.2 mg/L) (Figure 1B).

### 3.2. Composition and Ultrastructural Examination of Precipitates and Remineralisation Model

Raman analyses were performed to compare the chemical composition of the demineralised dentin, remineralised dentin through the experimental model, and control dentin. Raman spectra obtained from remineralised dentin at the end of the experiment (day 120) revealed distinctive apatite Raman bands at 955 and 1063 cm^−1^. In the remineralised dentin spectra, these Raman bands had a higher intensity than those of demineralised dentin and closer intensity to control dentin (Figure 1D). Moreover, demineralised dentin showed lower intensity Raman bands at 1240–1255 cm^−1^ (amide III from peptide bonds), 1454 cm^−1^ (CH2), and 1655 cm^−1^ (amide I from peptide bonds) regions, which are related to organic compounds (peptide backbone). However, these bands disappeared in the remineralised dentin spectra (Figure 1D).

Precipitates formed on the demineralised dentin matrix throughout the experiment were assessed by SEM-EDAX. Results showed that the morphological characteristics of precipitates and Ca/P molar ratios varied over time. Starting from day 1, spherule precipitates with acicular crystallites (Figure 1C) were formed on the demineralised dentin surface. Semi-quantitative analysis of the chemical composition of those precipitates indicated a Ca/P molar ratio of 1.44 and other elements in trace amounts (O, Si, Mg, Fe, and K). Petal-like precipitates (Figure 1C) were also observed in the samples starting from day 1, with a Ca/P molar ratio of 1.58. From day 15 onwards, compact, lath-like precipitates were observed (Figure 1C), with a Ca/P molar ratio of 1.68.

SEM analyses were performed to observe a more accurate description of the remineralisation process. Photomicrographs revealed the presence of precipitates with different morphologies over time, as described in the SEM-EDAX analyses. In the first week, mineral deposition was initiated, and the nucleation of calcium phosphate crystals with high affinity to bind intratubular dentin was observed (Figure 2). The deposition of precipitates increased over time. At 2 weeks, a compact mineralised layer was observed around the dentin tubules. At 1 month, several mineralisation nodes merged, forming integrated structures. At 2 months, the demineralised dentin matrix was covered by an uneven mineralised layer. By the end of the experiment (4 months), a complete mineralised layer covered the demineralised dentin matrix.

SEM analysis of the partially demineralised dentin surface showed permeable tubules in the demineralised area (Figure 3A,B). We observed an increase over time in the number of apatite-like clusters, which were deposited, mainly, over the surface of the demineralised dentin. Untreated dentin showed only a very small amount of mineral precipitation (Figure 3C–E). Over time, a more extensive remineralisation layer was observed. It is important to highlight that despite the increased precipitation amount and thickening of the mineralised layer, mineral deposition occurred only over the surface of demineralised dentin (Figure 3E,F). After 4 months, a complete mineralised layer was achieved (Figure 3G,H).

### 3.3. Functional Relationship between MMPs and NCPs

Zymography analyses were performed to observe the effect of demineralisation and remineralisation treatment on dentin MMPs activity. Additionally, samples were treated with chlorhexidine to inhibit the activity of MMPs used as a negative control (R1–R5). Zymography revealed gelatinolytic bands at 92, 68, and 54 kDa molecular weights, associated with MMP-9, MMP-2, and MMP-20, respectively. Untreated dentin (control) and demineralised dentin showed the presence of MMP-2, MMP-9, and MMP-20. Remineralised dentin showed bands associated with MMP-2 and MMP-20. No gelatinolytic activity was observed neither in G3 nor G5, indicating that the treatment was effective at inhibiting the activity of MMPs (Figure 4A).

To assess localisation of MMP-2 and MMP-9 in the experimental groups (G1–G5), immunohistochemistry was performed. Immunoreactivity analysis confirmed the presence of MMP-2 and MMP-9, specifically in the intratubular and intertubular dentin. Demineralised and remineralised dentin showed significantly higher levels of MMP-2 and MMP-9 compared to control (*p* < 0.05). Further, chlorhexidine-treated groups (demineralised and remineralised dentin) showed significantly lower levels of MMP-2 and MMP-9 compared to control (*p* < 0.01) (Figure 4B,C).

Using immunohistochemistry, DMP1-CT was observed mainly in intratubular dentin; however, DSPP was observed in intratubular and intertubular dentin. Demineralised and remineralised dentin groups showed a significant increase in protein levels compared with control (*p* < 0.05). Groups inhibited with chlorhexidine had significantly lower levels than the experimental groups (*p* < 0.05) (Figure 5).

DMP1-CT immunoreactivity was mainly localised in intratubular dentin; however, a few dentinal tubules did not contain DMP1-CT (Figure 6A). SEM photomicrographs showed that crystal nucleation and mineralised layer deposition started from intratubular dentin. Nevertheless, the mineralisation cluster deposition localised around the opening of the dentin tubules was not uniformly observed (Figure 6B–E). The immunohistochemical reaction for DSPP was very intense in some areas, which seemed to match with areas of higher mineral deposition, as observed in SEM analyses (Figure 6F,G).

## 4. Discussion

Our study was designed to assess the role of MMPs and NCPs as active promoters for the remineralisation of radicular dentin through a cell-free modified remineralisation model [27]. Using this model, we observed mineral precipitation on the demineralised dentin surface after the first 12 h, with an increase in mineral deposition over time. SEM photomicrographs showed crystal deposition singularly on the demineralised dentin surface on which NCPs were exposed, suggesting that, through the exposure of NCPs, specific binding sites guide the mineral deposition on the radicular dentin matrix.

We recorded an initial increase in pH and calcium-ion release (up to day 3). This increase could be due to the release of OH^−^ and Ca^2+^ ions from the bioactive material, which is related to the initial production of amorphous calcium phosphate, the first stage of carbonate apatite formation [35,36]. During the evaluation period, the pH and calcium profile showed a decrease, which became stable after day 15, and could be attributed to the incorporation of ions into the apatite in the second stage of carbonate apatite formation [10,27].

Our findings support a previous study, which stated that the Raman technique is useful to measure mineral loss during demineralisation procedures [37]. Thus, we used the Raman technique to evaluate remineralisation in our experimental model. We observed predominant bands at 955 and 1063 cm^−1^, which could be attributed to the bending vibration of ν1 (PO_4_^3−^) within hydroxyapatite crystals [37,38]. These peaks showed lower intensity in demineralised dentin compared to control dentin, suggesting an important decrease in the presence of mineral compounds. Remineralised dentin showed a higher intensity of these bands when compared to demineralised dentin. Additionally, we identified Raman bands at 1454 cm^−1^, which were tentatively assigned to CH_2_ deformation of the matrix collagen [37,38]. These bands seemed to disappear in the Raman spectra of remineralised dentin when compared to demineralised dentin, suggesting that our model achieved complete deposition of the mineral layer (remineralisation) over the collagen matrix.

Ultrastructural examination of the precipitates and remineralised layer allowed observation of the timing and patterns of remineralisation. Our study developed an experimental model in which dentin, the tissue to remineralise, was not in direct contact with the bioactive material. We observed that precipitation over the PBS fluid began in the first hour after immersion, and an increase was observed throughout the experimental period. For the first time, our results demonstrated migration of the mineral precipitates from their ion source (bioactive material) to dentin, indicating the presence of an attraction flux and specific sites of nucleation. We observed spherule precipitates with acicular crystals (Ca/P: 1.44) after day 1, which could be related to the transformation of the metastable amorphous calcium phosphate phase into the apatite phase [27]. Additionally, petal-like (Ca/P: 1.58) and compact lath-like precipitates (Ca/P: 1.68) were identified after one week. The petal-like precipitates are likely to represent octocalcium phosphate, a mandatory precursor for the formation of carbonated apatite [10,25]. The compact lath-like structures are compatible with the formation of carbonated apatite, as suggested in previous studies [35,36,39]. Interestingly, the three forms of precipitates were observed after one week and throughout all experimental time periods in our model, suggesting a renovated and continuous biomineralisation process over time.

Furthermore, ultrastructural examination showed that crystal deposition occurred mainly on the previously demineralised surface, and not on the untreated surface or control dentin. Crystal nucleation was initially observed in the intratubular zone of the demineralised radicular dentin. We believe that this phenomenon occurs because the dentinal collagen matrix is a NCPs reservoir, which functions as a template for mineral nucleation, followed by cluster formation and subsequently results in complete remineralisation. The demineralisation procedure exposed specific phosphoric acid-containing signalling molecules such as DMP1-CT and DSPP within the underlying dentin layer, initiating the specific binding of mineral precipitates to specific sites of dentin and triggering the remineralisation process.

Immunohistochemistry showed increased levels of both DMP1 and DSPP in demineralised radicular dentin compared to control dentin. DMP1 and DSPP have previously been described as active promoters and controllers of the biomineralisation of dentin [5,10,40]. Indeed, previous reports have provided solid evidence that DSPP and its cleaved products are critical for dentin mineralisation, and they may function synergistically with DMP1 and the cleaved by-products of both proteins [40]. The acidic nature of DMP1 is related to a high calcium ion-binding capacity, which is essential for its important role in mineralisation [5]. Moreover, compelling evidence suggests that full-length DMP1 is a precursor and needs to be cleaved to generate C-terminal and N-terminal functional forms [41].

Gelatine zymography and immunohistochemistry also revealed a higher presence of MMP-2 and MMP-9 in demineralised dentin than in control dentin. It is widely accepted that MMPs convert structural matrix proteins into signalling molecules, generating peptides with specific biological activities [15]. DMP1 could be cleaved by MMP-2 to release a C-terminal peptide. Due to its location in mineralised matrices, it could be a nucleator of mineralisation [15]. It is important to highlight that our experimental model used a distance of 15 mm between the dentin slice and bioactive material. Consequently, the calcium ions and apatites had to be mobilised at least 15 mm to reach the exposed dentin matrix layer. Thus, after demineralisation of radicular dentin, these highly phosphorylated fragments, such as DMP1-CT, became exposed. It is likely that after the demineralisation procedure, MMP-2 and MMP-9 became active and cleaved NCPs, such as DMP1, to release the C-terminal peptide, which is suitable for guiding the recruitment and binding of calcium ions [10]. Our findings suggest that this process allows a specific attraction flux of apatite through the PBS solution to remineralise the demineralised layer throughout the surface.

Immunohistochemistry showed the presence of DMP1-CT predominantly in intratubular demineralised dentin. Indeed, we observed the initial cluster formation emerging from the interior of the dentinal tubules. Thus, we speculate that there is a functional relationship between the exposed DMP1-CT and mineral deposition, which was reinforced by the lack of precipitation on the surface of control dentin. Mineral precipitation occurred mainly inside or around dentinal tubules, suggesting that the presence of NCPs, specifically the C-terminal of DMP1, is necessary for the specific ion-binding biomineralisation of the tissue.

The immunoreaction for DMP1-CT and DSPP was not homogeneous throughout radicular dentin samples. Some dentinal tubules without detectable DMP1-CT and DSPP and areas with increased levels of these proteins were observed. Furthermore, SEM photomicrographs revealed a similar pattern, with cluster deposition related to dentinal tubules, but with some areas with no precipitates and others with larger amounts of precipitates, creating an irregular mineralised layer. A possible explanation for this distribution is that NCPs could be proteolytically degraded by previous clinical conditions, such as exodontics, inflammation, heat, or necrotic pulp. This potential requirement for active NCPs, therefore, highlights a clinical challenge. Nevertheless, it may be possible to use biomimetic analogues of phosphoproteins to re-establish the biological characteristics of the ECM [42]. However, the mechanisms underlying the function of biomimetic analogues and their actual ensembles in the dentin matrix remain a matter of debate, since the analogues cannot be proven to bind to the specific location, emulating the complex structure of the dentin layer.

Notably, gelatine zymography showed no gelatinolytic activity in the groups treated with chlorhexidine, an effective MMP-2 and MMP-9 inhibitor [32,43,44]. This finding was corroborated by immunohistochemistry assays for MMP-2 and MMP-9, which revealed significantly lower—almost undetectable—expression in these samples. Moreover, DMP1-CT and DSPP levels were similarly reduced in these groups. Studies have suggested the use of chlorhexidine as an endodontic irrigant due to its antimicrobial and substantive properties and its ability to inhibit MMP-mediated degradation of the ECM, which would improve adhesion of resin-based materials [32,43,44]. However, it is important to highlight, based on our results, the use of chlorhexidine may not be suitable in cases in which dentin matrix remineralisation and regeneration is desired. Inhibition of MMP activity could compromise the activation of signalling molecules (e.g., DMP1-CT) and regeneration of the mineral layer.

In summary, our study suggest that the remineralisation of radicular dentin can be achieved with the aid of bioactive materials and a standardised demineralisation procedure to expose NCPs in the dentin matrix, allowing MMPs to convert structural matrix proteins into signalling molecules and generating peptides, which allows a specific attraction flux of apatites through the PBS solution, to guide mineral nucleation and maturation. From a clinical perspective, it is important to state that the dentinal matrix possesses its own biological resources to regenerate, and clinicians must make an effort in preserving NCPs when dentin remineralisation and/or tissue regeneration are to be achieved. However, in cases in which the dentin matrix is damaged due to several pathological conditions with evident protein degradation, a different strategy with molecular signalling analogues might be required.

Thus, ensuring the mimicking of natural radicular dentin tissue with molecular NCPs and collagen matrix analogues should be a focus of future research.

## Figures and Tables

**Figure 1 materials-13-01053-f001:**
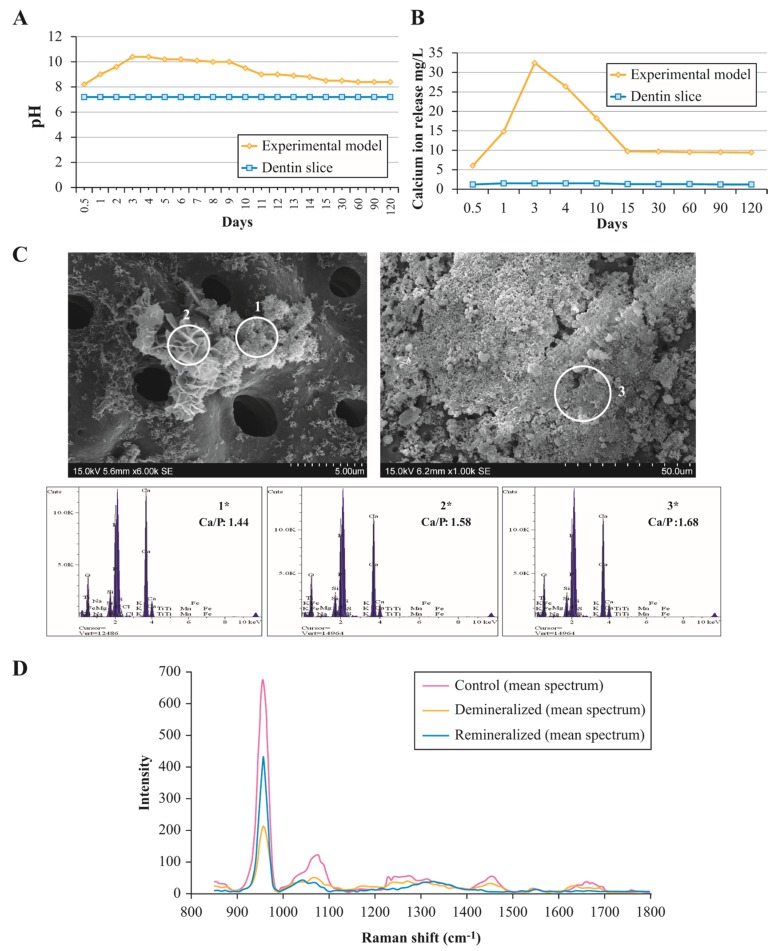
Composition and ultrastructure evaluation of biomimetic remineralisation system. (**A**) pH profiles of solutions. (**B**) Calcium ion release profile. (**C**) Morphological characterisation of precipitates by scanning electron microscopy (SEM). (1) Energy dispersive X-ray analysis (EDAX) spectrum for acicular precipitates and semi-quantitative chemical composition showing a Ca/P molar ratio of 1.44, which could be related to the transformation of the metastable amorphous calcium phosphate phase into the apatite phase. (2) Petal-like precipitates (6000×) and EDAX, revealing a greater Ca/P molar ratio of 1.58. The petal-like precipitates are likely to represent octocalcium phosphate, a mandatory precursor for the formation of carbonated apatite. (3) Compact lath-like precipitates (1000×) and EDAX, revealing a greater Ca/P molar ratio of 1.68, compatible to carbonated apatite. (**D**) Micro-Raman mean spectra obtained from control dentin, demineralised dentin, and remineralised dentin. The predominant bands at 955 and 1063 cm^−1^ attributed to the bending vibration of ν1 (PO_4_^3−^) within hydroxyapatite crystals denote a lower intensity in remineralised dentin and an even lower intensity in demineralised dentin compared with untreated dentin. Raman bands at 1454 cm^−1^ assigned to CH_2_ deformation of the matrix collagen seem to disappear in the spectra of remineralised dentin compared to demineralised dentin, suggesting that our model achieved complete deposition of the mineral layer (remineralisation) over the collagen matrix.

**Figure 2 materials-13-01053-f002:**
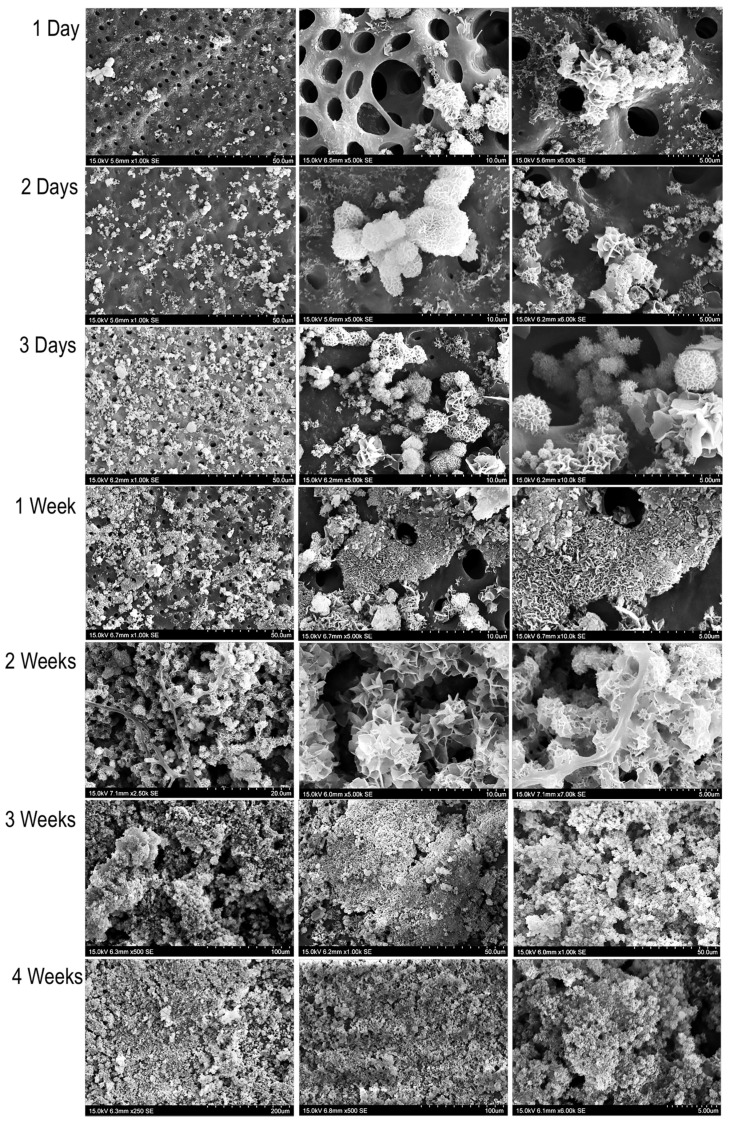
Evaluation of remineralisation by the biomimetic mineralisation model. Scanning electron microscopy (SEM) analysis of mineral deposition on demineralised dentin at the different periods of time. Three levels of magnification are shown for each timepoint. Mineral precipitation was observed mainly in the interior and surrounding the dentinal tubules. Over time, it was noted that our experimental model promoted remineralisation of the demineralised radicular dentin.

**Figure 3 materials-13-01053-f003:**
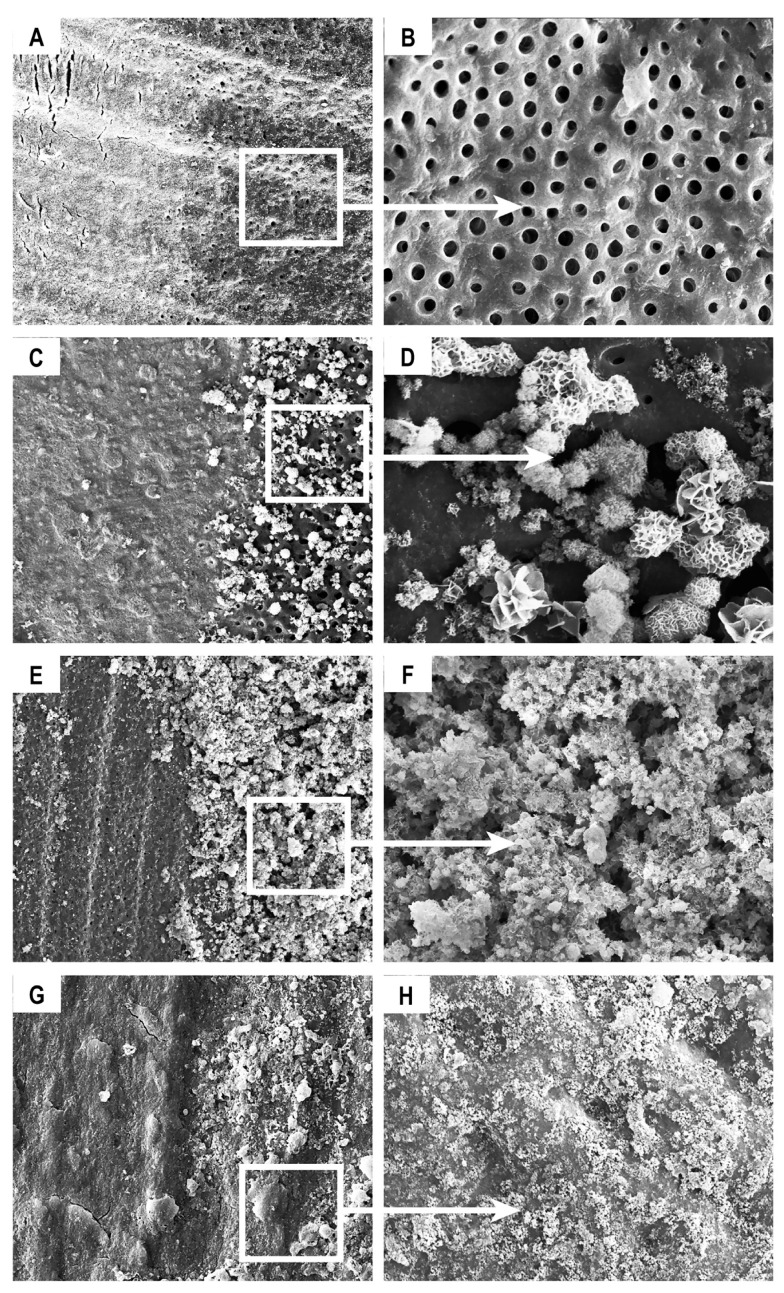
Photomicrographs of partially demineralised dentin as a template for remineralisation. Scanning electron microscopy (SEM) analysis showed (**A**) partially demineralised dentin (500×) and (**B**) permeable tubules (2000×); (**C**) precipitate formation exclusively over the demineralised dentin surface (500×) and (**D**) inside the intratubular dentin area (5000×); **E** (250×), **F** (2000×) thickening of the mineralised layer exclusively over the demineralised dentin at day 60; and **G** (2000×), **H** (5000×) remineralisation of radicular dentin was observed (4 months). Control dentin is shown on the left-hand side of panels a, c, e, and g. Uncropped SEM images in Appendix A.

**Figure 4 materials-13-01053-f004:**
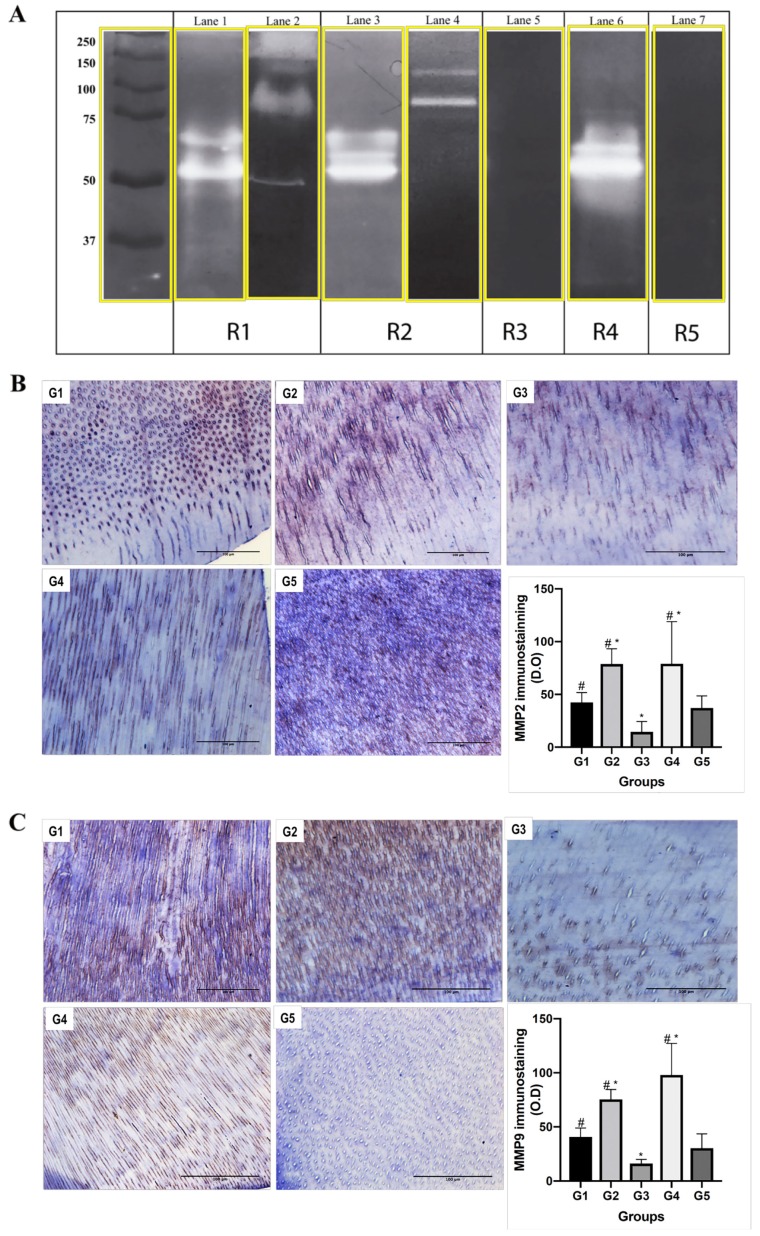
Presence of matrix metalloproteinases. (**A**) Representative lanes of different zymography gels, showing the presence of gelatinolytic bands at 54 (MMP-20), 68 (MMP-2), and 92 kDa (MMP-9). Uncropped zymograms in Appendix A. (**B**) Representative images of MMP-2 immunohistochemical analysis (40×). Scale bars, 100 μm. (**C**) Representative images of MMP-9 immunohistochemical analysis (40×). Scale bars, 100 μm. Immunostaining was observed mainly in the intratubular dentin. Samples were treated with chlorhexidine to inhibit the activity of MMPs and used as a negative control (G5). At the right bottom of b and c, immunostaining is expressed as optical density (OD). Each bar represents the mean ± standard error of the mean. * *p* < 0.05 versus G1 (control group). # *p* < 0.05 versus G3.

**Figure 5 materials-13-01053-f005:**
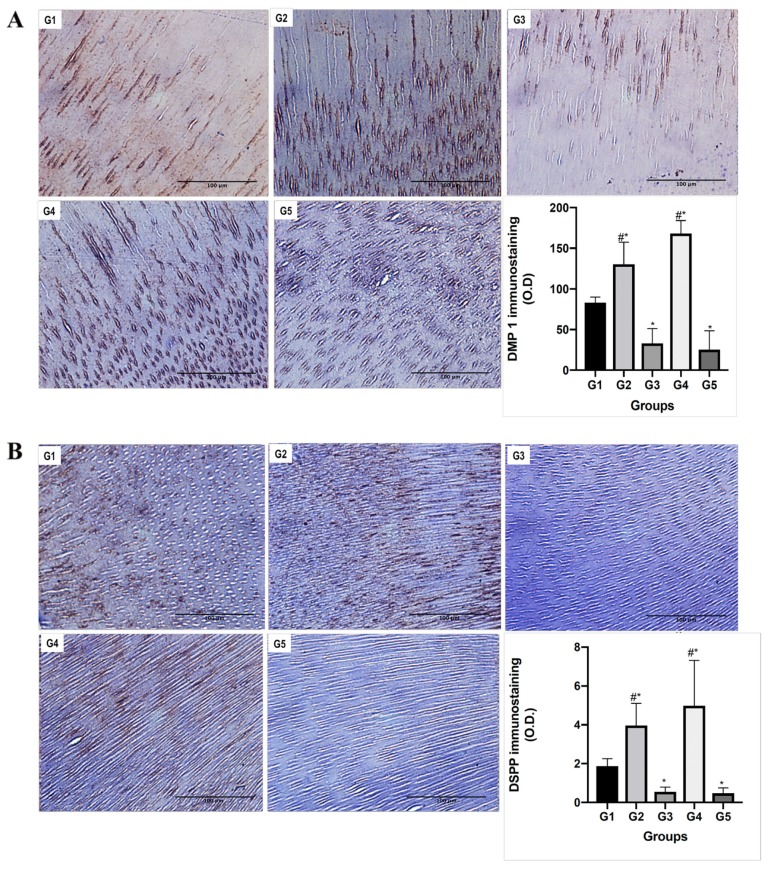
DMP1-CT and DSPP immunoreactivity. (**A**) Representative images for DMP1-CT and (**B**) DSPP immunohistochemical analyses. Scale bars, 100 μm. DMP1-CT was observed mainly in intratubular dentin; however, DSPP was observed in intratubular and intertubular dentin. Samples treated with chlorhexidine were used as a negative control (G5). At the bottom right of each panel, immunostaining is expressed as optical density (OD). Each bar represents the mean ± standard error of the mean. * *p* < 0.05 versus G1 (control group). # *p* < 0.05 versus G3.

**Figure 6 materials-13-01053-f006:**
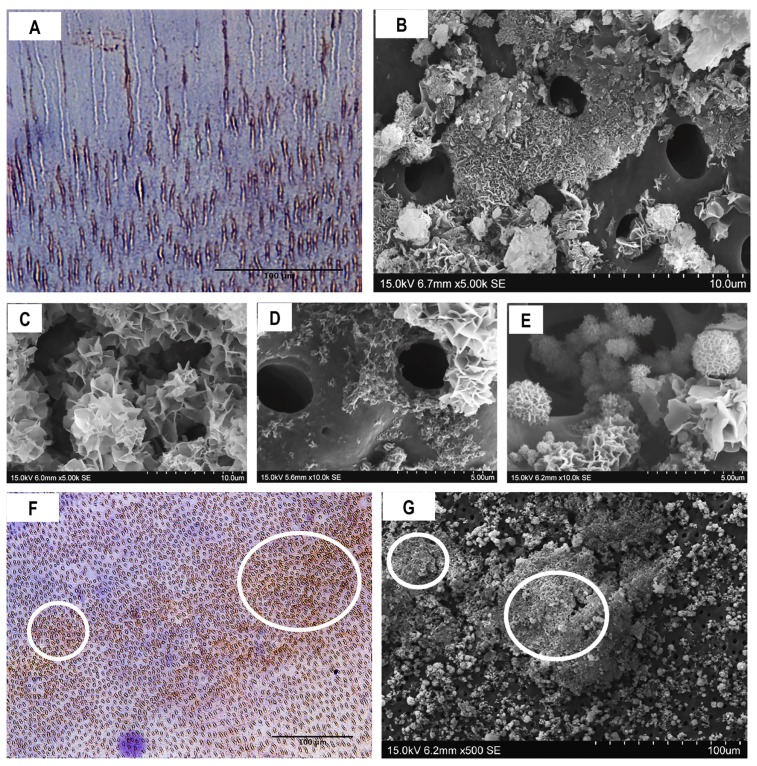
Functional relationship between NCPs and remineralisation of radicular dentin. (**A**) DMP1-CT immunoreactivity (40×). Scale bars, 100 μm. (**B**–**E**) scanning electron microscopy (SEM) photomicrographs showing mineral formation at intratubular and peritubular dentin at early stages. (**F**) DSPP immunoreactivity (10×). Circles indicate areas of greater immunoreactivity, and (**G**) the SEM image shows similar higher mineral deposition in some areas. Circles indicate areas with high mineral deposition.

**Table 1 materials-13-01053-t001:** Distribution of experimental groups.

Groups	Treatments
G1	Control group
G2	37% H_3_PO_4_ for 30 s + distilled water
G3	37% H_3_PO_4_ for 30 s + distilled water + 5 min in 2% chlorhexidine + distilled water
G4	37% H_3_PO_4_ for 30 s + distilled water + Biomimetic remineralisation model
G5	37% H_3_PO_4_ for 30 s + distilled water + 5 min in 2% chlorhexidine + Biomimetic remineralisation model

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
