# Peer review of "Non-Collagenous Dentin Protein Binding Sites Control Mineral Formation during the Biomineralisation Process in Radicular Dentin"

_materials, 2020, doi:10.3390/ma13051053_

Round 1

Reviewer 1 Report

The authors are to be congratulated by this very interesting and well written paper.

The only issue I should raise regards the somewhat inconsistent figure formatting. This is indeed a very minor point, and entirely up to the authors to follow, but I would suggest a more consistent and careful assembly of the various panels would benefit the reading. For instance:

1) Figure 2, instead of A-G entries, add the time point on the left of each row, and the "magnification" on top of each column, and crop the repeating SEM conditions from the images and add this information to the legend. Furthermore, add scale bars to each image.

2) Figure 3: use higher contrast letters (white, or black on white boxes). Also, add scale bars to each image.

3) Figure 4 and 5: move images to G1-3 (upper) and G4-5 (lower row); also, move letters to top-left corner, and add scale bars to each image. Also, adjust bars in graphs to avoid over position of symbols.

4) Figure 6: similar as described above for either SEM or light microscopy images.

Author Response

Responses to Reviewers Manuscript ID:  materials-724163.

Title: Non-collagenous dentin proteins binding sites controls mineral formation during the biomineralisation process in radicular dentin.

We are truly grateful to the reviewers for their constructive comments. We have amended all minor suggestions, clarified the unclear parts and the manuscript has been revised accordingly. Our responses to the queries raised are as follows (responses in bold):

Reviewer #1:

The authors are to be congratulated by this very interesting and well-written paper.

We are truly grateful for your constructive comments.  

The only issue I should raise regards the somewhat inconsistent figure formatting. This is indeed a very minor point, and entirely up to the authors to follow, but I would suggest a more consistent and careful assembly of the various panels would benefit the reading. For instance:

1) Figure 2, instead of A-G entries, add the time point on the left of each row, and the "magnification" on top of each column, and crop the repeating SEM conditions from the images and add this information to the legend. Furthermore, add scale bars to each image.

Figure 2 was amended, and the A-G entries were eliminated. Moreover the time points were added. However, we must clarified that the scale bar of each SEM image is in the same “black box” were the SEM conditions can be observed. The scale bar is at the right lower corner of each SEM image.

2) Figure 3: use higher contrast letters (white, or black on white boxes). Also, add scale bars to each image.

We added white boxes with black letters. SEM Photomicrographs were cropped and rotated to allow a better visual comparison, showing control dentin at the left-hand side. Thus, we added the uncropped SEM photomicrographs in the Supplementary information as Figure S2.

3) Figure 4 and 5: move images to G1-3 (upper) and G4-5 (lower row); also, move letters to top-left corner, and add scale bars to each image. Also, adjust bars in graphs to avoid over position of symbols.

Figures 4 and 5 were amended accordingly your suggestions. The letters were moved to the top-left corner and the scale bars added to each image.

4) Figure 6: similar as described above for either SEM or light microscopy images.

Figure 6 were amended accordingly your suggestions. SEM images have the scale bar at the right lower corner.

Reviewer 2 Report

Retana-Lobo et al assessed the functional relationship of metalloproteinases (MMPs: MMP-2 and MMP-9) and non-collagenous proteins (NCPs: DSPP and DMP1-CT) with mineral initiation and maturation during the biomineralization of radicular dentin. They observed that mineral nucleation and growth occurred on the demineralized radicular dentin surface. Also, carbonate apatite formed during the remineralization of dentin. DMP1-CT and DSPP binding sites controls the carbonate apatite nucleation and maturation guiding the remineralization of radicular dentin.

This work is generally well organized and executed. Additionally, the manuscript is quite well written with clear flow. Overall, this study is interesting and presents new insights into the role of MMPs and NCPs in the process of mineralization of radicular dentin and should attract the interest of the readers of Materials. Some minor issues, however, need to be addressed to improve the quality of the current work.

There is no scale bar in Figure 3. Please provide. Similarly, there is no scale bar in Figure 4B, C, Figure 5A, B, and Figure 6A, F.

Author Response

Responses to Reviewers Manuscript ID:  materials-724163.

Title: Non-collagenous dentin proteins binding sites controls mineral formation during the biomineralisation process in radicular dentin.

We are truly grateful to the reviewers for their constructive comments. We have amended the minor suggestions, clarified the unclear parts and the manuscript has been revised accordingly. Our responses to the queries raised are as follows (responses in bold):

Reviewer #2:

Retana-Lobo et al assessed the functional relationship of metalloproteinases (MMPs: MMP-2 and MMP-9) and non-collagenous proteins (NCPs: DSPP and DMP1-CT) with mineral initiation and maturation during the biomineralization of radicular dentin. They observed that mineral nucleation and growth occurred on the demineralized radicular dentin surface. Also, carbonate apatite formed during the remineralization of dentin. DMP1-CT and DSPP binding sites controls the carbonate apatite nucleation and maturation guiding the remineralization of radicular dentin.

This work is generally well organized and executed. Additionally, the manuscript is quite well written with clear flow. Overall, this study is interesting and presents new insights into the role of MMPs and NCPs in the process of mineralization of radicular dentin and should attract the interest of the readers of Materials.

We are truly grateful for your constructive comments.  

Some minor issues, however, need to be addressed to improve the quality of the current work.

There is no scale bar in Figure 3. Please provide. Similarly, there is no scale bar in Figure 4B, C, Figure 5A, B, and Figure 6A, F. 

Figures 3, 4, 5 and 6 were amended accordingly your suggestions. The letters were moved to the top-left corner and the scale bars added to each image.

Reviewer 3 Report

Review materials-724163

Non-collagenous dentin proteins binding sites 3 controls mineral formation during the 4 biomineralisation process in radicular dentin.

Reviewer comments:

This research has a correct design, and the paper is well written. However, minimal corrections can help the reader and improve the paper.

Introduction, page 2 line 52:

Please, define RGD the first time this abbreviation appears in the text.

Materials and methods, page 3 line 120:

Provide the sample size calculation according to the main objective of the study.

Materials and methods, page 3 line 138:

Please, define PBS the first time this abbreviation appears in the text, and provide the manufacturer if applicable.

Author Response

Responses to Reviewers Manuscript ID:  materials-724163.

Title: Non-collagenous dentin proteins binding sites controls mineral formation during the biomineralisation process in radicular dentin.

We are truly grateful to the reviewers for their constructive comments. We have amended the minor suggestions, clarified the unclear parts and the manuscript has been revised accordingly. Our responses to the queries raised are as follows (responses in bold):

Reviewer #3:

Reviewer comments:

This research has a correct design, and the paper is well written.

We are truly grateful for your constructive comments.

However, minimal corrections can help the reader and improve the paper.

Introduction, page 2 line 52:

Please, define RGD the first time this abbreviation appears in the text.

The text was amended in page 2 - line  52. “…,and the Arg-Gly-Asp (RGD) cell-binding sequence…”

Materials and methods, page 3 line 120:

Provide the sample size calculation according to the main objective of the study.

The main objective of our study was to evaluate the functional relationship of MMPs (MMP-2 and MMP-9) and NCPs (DSPP and DMP1-CT) with the initiation and maturation of apatite, through a biomimetic remineralisation system of radicular dentin. The hypothesis was that NCPs binding sites acts as mineral nucleators guiding the remineralisation of radicular dentincollagen matrix with the aid of bioactive materials. Thus, our main objective was evaluated in a descriptive and observational manner in SEM evaluation, zymography and immunohistochemical expression.  

Our study performed an ex vivo biomimetic remineralisation model with controlled variables. A flowchart of study samples selection was included in the supplementary information as Figure S1. In the immunohistochemical statistical analyses the confidence level was of 95% and the sample size was based on previous studies.  Some examples:

*Reyes-Carmona, J.F.; Felippe, M.S.; Felippe, W.T. A phosphate-buffered saline intracanal dressing improves the biomineralization ability of mineral trioxide aggregate apical plugs. J. Endod. 2010, 36, 1648–1652; doi:10.1016/j.joen.2010.06.014.

*Reyes-Carmona, J.F.; Felippe, M.S.; Felippe, W.T. Biomineralization ability and interaction of mineral trioxide aggregate and white portland cement with dentin in a phosphate-containing fluid. J. Endod. 2009, 35, 731–736; doi:10.1016/j.joen.2009.02.011.

*Dreger, Luonothar Antunes Schmitt et al.Mineral Trioxide Aggregate and Portland Cement Promote Biomineralization In Vivo. Journal of Endodontics, Volume 38, Issue 3, 324 – 329.

*Reyes-Carmona, Jessie F. et al.  In Vivo Host Interactions with Mineral Trioxide Aggregate and Calcium Hydroxide: Inflammatory Molecular Signaling Assessment. Journal of Endodontics, Volume 37, Issue 9, 1225 – 1235.

Materials and methods, page 3 line 138:

Please, define PBS the first time this abbreviation appears in the text, and provide the manufacturer if applicable.

The text was amended in page 4 - line  138- 140. “A calcium-free and magnesium-freePhosphate Buffered Saline (PBS) solution containing 136.4 mM NaCl, 2.7 mM KCl, 8.2 mM NaH2PO4, and 1.25 mM KH2PO4in deionized water (pH 7.2) was prepared and filtered.”

We hope that we have responded to the Reviewers´ minor suggestions satisfactorily and have improved the manuscript in accordance with the criteria for publication in Materials.

Again, we thank the Reviewers for their very useful advice and suggestions.